

# Sesamoids in Caudata and Gymnophiona (Lissamphibia): absences and evidence

María Laura Ponssa[1] and Virginia Abdala[2]

[1] Área Herpetología, Unidad Ejecutora Lillo (UEL), CONICET-Fundación Miguel Lillo, San Miguel de Tucumán, Tucumán, Argentina
[2] Instituto de Biodiversidad Neotropical (IBN), UNT-CONICET. Cátedra de Biología General, Facultad de Ciencias Naturales e IML, UNT, Yerba Buena, Tucuman, Argentina

Corresponding author
María Laura Ponssa,
mlponssa@hotmail.com

## ABSTRACT

An integrative definition of sesamoid bones has been recently proposed, highlighting their relationship with tendons and ligaments, their genetic origin, the influence of epigenetic stimuli on their development, and their variable tissue composition. Sesamoid bones occur mainly associated with a large number of mobile joints in vertebrates, most commonly in the postcranium. Here, we present a survey of the distribution pattern of sesamoids in 256 taxa of Caudata and Gymnophiona and 24 taxa of temnospondyls and lepospondyls, based on dissections, high-resolution X-ray computed tomography from digital databases and literature data. These groups have a pivotal role in the interpretation of the evolution of sesamoids in Lissamphibia and tetrapods in general. Our main goals were: (1) to contribute to the knowledge of the comparative anatomy of sesamoids in Lissamphibia; (2) to assess the evolutionary history of selected sesamoids. We formally studied the evolution of the observed sesamoids by optimizing them in the most accepted phylogeny of the group. We identified only three bony or cartilaginous sesamoids in Caudata: the mandibular sesamoid, which is adjacent to the jaw articulation; one located on the mandibular symphysis; and one located in the posterior end of the maxilla. We did not observe any cartilaginous or osseous sesamoid in Gymnophiona. Mapping analyses of the sesamoid dataset of urodeles onto the phylogeny revealed that the very conspicuous sesamoid in the mandibular symphysis of *Necturus beyeri* and *Amphiuma tridactylum* is an independent acquisition of these taxa. On the contrary, the sesamoid located between the maxilla and the lower jaw is a new synapomorphy that supports the node of *Hydromantes platycephalus* and *Karsenia coreana*. The absence of a mandibular sesamoid is plesiomorphic to Caudata, whereas it is convergent in seven different families. The absence of postcranial sesamoids in salamanders might reveal a paedomorphic pattern that would be visible in their limb joints.

## INTRODUCTION

In a recent review, sesamoids were defined as "*periarticular skeletal elements, which initially form in juxtaposition to or independently of bones and joints. They are commonly related to tendons and ligaments, have a genetic basis and, once they are formed, epigenetic*
stimuli drive their growth and development to the acquisition of their definitive tissue composition, which can be diverse, for example, cartilage, fibrocartilage, or bone" (*Abdala et al., 2019*). This definition intends to summarize all the skeletal elements considered as sesamoids in the different groups of tetrapods into one description. The definition is quite broad, because sesamoids are associated with diverse functions, such as the mechanical stress exerted on tendons as they wrap around a bony edge or a joint. This stress might improve tendon ability to react to compressive load, pressure, tensile strain, or vibration (*Pearson & Davin, 1921a*, *1921b*; *Carlsöö, 1982*; *Nussbaum, 1982*; *Sarin et al., 1999*; *Benjamin & Ralphs, 1998*; *Jerez, Mangione & Abdala, 2010*; *Ponssa, Goldberg & Abdala, 2010*; *Tsai & Holliday, 2011*; *Otero & Hoyos, 2013*; *Regnault, Hutchinson & Jones, 2016*; *Abdala, Vera & Ponssa, 2017*; *Zhang et al., 2017*). Moreover, sesamoids are described in all the large vertebrate groups (*Abdala et al., 2019*). They occur mainly associated with a large number of mobile joints, most commonly in the postcranium (*Romankowowa, 1961*; *Vickaryous & Olson, 2007*; *Ponssa, Goldberg & Abdala, 2010*; *Jerez, Mangione & Abdala, 2010*; *Chadwick et al., 2014*; *Regnault, Pitsillides & Hutchinson, 2014*; *Reyes-Amaya, Jerez & Flores, 2017*; *Samuels, Regnault & Hutchinson, 2017*; *Denyer, Regnault & Hutchinson, 2020*), and in the skull of some taxa (*Hofling & Gasc, 1984*; *Tsai & Holliday, 2011*; *Montero et al., 2017*). The number of cranial sesamoids is notoriously higher in Osteichthyes than in tetrapods (*Alexander, 1967*; *Adriaens & Verraes, 1998*; *Diogo, Oliveira & Chardon, 2001*; *Summers et al., 2003*; *Datovo & Bockmann, 2010*), possibly due to the higher number of mobile joints in the fish skull (*Iordansky, 1989*; *Montero et al., 2017*; *Abdala et al., 2019*).

More than 20 types of osseous, cartilaginous or fibrocartilaginous sesamoids have been described in amphibians (*Laurent, 1961*; *Nussbaum, 1982*; *Olson, 2000*; *Hoyos, 2003*; *Guayasamin, 2004*; *Avilan & Hoyos, 2006*; *Fabrezi, 2006*; *Ponssa, Goldberg & Abdala, 2010*; *Abdala, Vera & Ponssa, 2017*), most of them in 33 extant families of anurans (*Ponssa, Goldberg & Abdala, 2010*). The earliest records of sesamoids in amphibians include an oval bone lying next to the radial, and distal to the ulnar portion of the radio-ulna in *Xenopus arabiensis* from the Upper Oligocene (*Henrici & Baéz, 2001*). This element is located on the dorsal surface of the carpus in the same position as in other *Xenopus* species (*Henrici & Baéz, 2001*). A sesamoid behind the diapophysis, beneath the iliac shaft, was identified in a juvenile anuran from the Lower Cretaceous Jiufotang Formation (*Wang, Jones & Evans, 2007*).

The skeletal anatomy of Caudata and Gymnphiona has been extensively studied (*Wake, 1963*; *Bemis, Schenk & Wake, 1983*; *Wake, Wake & Wake, 1983*; *Restrepo, 1995*; *Shubin, Wake & Crawford, 1995*; *Müller, 2006*; *Schaaf, 2010*; *Wu, Wang & Hanken, 2012*; *Villa et al., 2014*; *Jia et al., 2018*; *Cala, Tarazona & Ramírez-Pinilla, 2019*; *Khoshnamvand et al., 2019*; *Bardua et al., 2019*; *Marshall et al., 2019*; *Macaluso et al., 2020*). Nevertheless, no work has focused on their sesamoids. This gap of knowledge has attracted our attention, since the Caudata group has a pivotal role in the interpretation of the evolution of sesamoids in tetrapods. Likewise, Gymnophiona cannot be ignored if the evolution of sesamoid in amphibians is to be unveiled.

In Caudata, only a cranial sesamoid in the squamosal-collumelar ligament was mentioned (*Nussbaum, 1982*), with no data or information about the species having been published. Interestingly, cranial sesamoids have been rarely recorded among tetrapods (*Bramble, 1974*; *Tsai & Holliday, 2011*; *Montero et al., 2017*; *Abdala et al., 2019*). The sesamoids located in the cranium of tetrapod amniotes include the transilient cartilage or bone in the bodenaponeurosis of the jaw adductor muscles in crocodiles and turtles (*Iordansky, 1964*; *Schumacher, 1973*; *Holliday & Witmer, 2007*; *Tsai & Holliday, 2011*); elements in the basipterygoid and pterygoid bones in some squamates (*Gauthier et al., 2012*); the quadrate element in Ophiodes; the X element in amphisbaenians (*Montero et al., 2017*); a small bone strengthening the external jugo-mandibular ligament in its retro-articular portion in *Rhamphastos* (*Hofling & Gasc, 1984*); and the controversial Paaw cartilage in marsupial mammals (*Sánchez-Villagra et al., 2002*). The only records of cranial sesamoids in anuran amphibians belong to *Barbourula busuangensis* (*Clarke, 1987*; *Roček et al., 2016*) and *Chacophrys pierotti* (*Fabrezi, Goldberg & Chuliver Pereyra, 2017*). Considering these data, the report of the sesamoid in the columellar squamous ligament of Caudata is striking, given the notable scarcity of sesamoid records in this area for the entire tetrapod clade. In Gymnophiona, sesamoids are unknown. Data scarcity on sesamoid occurrence in Caudata and Gymnophiona draws our attention, because osteology is considered in several ongoing projects about the morphology of these taxa (*Bemis, Schenk & Wake, 1983*; *Müller, 2006*; *Villa et al., 2014*; *Bardua et al., 2019*; *Marshall et al., 2019*; among others).

To evaluate sesamoid evolution in these amphibian groups, and to fill the knowledge gap about them, we present a sesamoid distribution survey. Our main goals were:

1. to contribute to the knowledge of the comparative anatomy of bony or cartilaginous sesamoids in Lissamphibia;
2. to infer the evolutionary history of selected sesamoids in Lissamphibia.

Based on the available evidence of distribution patterns of bony and cartilaginous sesamoids in the remaining tetrapods, we consider that the lack of sesamoid records in urodeles and caecilians is probably due to sampling error. We expect to find bony or cartilaginous sesamoids in both groups: cranial sesamoids in Gymnophiona-due to its lack of limbs-and cranial and postcranial sesamoids in Caudata.

To achieve our goals, we studied about 850 specimens belonging to 280 taxa from dissected specimens, scanned specimens from different digital morphological databases, and literature data.

## MATERIALS AND METHODS

Sesamoids were examined in the skeleton of specimens from the herpetological collections of the Field Museum of Natural History, USA (FMNH) and Fundación Miguel Lillo, Argentina (FML). Material from three morphological databases was also reviewed: Morphosource (morphosource.org), Digimorph (http://digimorph.org/) and Phenome10k (phenome10k.org) (Supplemental Material S1). Adult and juvenile specimens representing all nine extant families (and one extinct family), 73 genera and 212 species (including

12 fossil species) of Caudata (571 specimens); nine (one extinct) of 11 families, 29 genera and 44 species (including one fossil species) of Gymnophiona (173 specimens) and 24 species of Temnospondyli and Lepospondyli (17 families) (Supplemental Material S1) were studied in detail. Dry skeletons and cleared and stained specimens of the collections were examined under a Meiji EMZ-5 binocular microscope. Clearing and staining allowed us to distinguish bony and cartilaginous structures. Unfortunately, since specimens were from the skeleton collection, it was not possible to determine age specifications, fixing conditions or loss of bones with soft tissue in the samples. However, our data on high-resolution X-ray computed tomography of specimens from digital databases allowed us to observe the internal structure of organisms without damaging the specimens, thus avoiding the problem of loss of bones with soft tissue.

Two individuals of *Pleuredeles waltl* were dissected to corroborate the identity of the ligaments or tendons related to the sesamoids. One specimen (FML30803) was completely cleared, while the other (FML30804) was partially cleared, and both were stained following the protocol of *Wassersug (1976)*, with modifications. Furthermore, some specimens in the FMNH collection were partially transparent (e.g., *Ambystoma mexicanum* FMNH22888), allowing the identification of tendons and associated muscles. Photographs were taken with a Nikon Coolpix P6000 camera, and with a Leica, MZ7.5 stereomicroscope equipped with a Spot Insight Color Model# 3.2.0 camera. The selected specimens were photographed with a DSLR camera, illuminated by light in the blue spectrum. The photos were taken through a yellow long-pass filter (which filters out any reflected light source, allowing only the fluorescent light to pass through). This visualization method of cleared and stained specimens was first described by *Smith et al. (2018)*. To compile the sesamoids described for urodeles and caecilians, the literature on myology and osteology of these groups was reviewed.

Descriptions of 66 species of Caudata and 40 species of Gymnophiona were also considered (see species and literature cited in Supplemental Material S1). The presence or absence of bony or cartilaginous sesamoids in each of the 280 studied taxa was recorded. Based on both our survey and the literature, the presence, number, and type of bony or cartilaginous sesamoids in urodeles and caecilians were inferred. These relationships were expressed through a probability calculation (Table 1). Our survey included extant species, although descriptions of fossil species were also revised (Supplemental Material S1). Because heterotopic elements are often overlooked in morphological studies, and several of the reviewed studies are descriptions of the skull only, results from literature sources (Supplemental Material S1) should be interpreted with caution. The topology of a suite of anatomical characters (skeletal elements, tendons, ligaments, and muscles) as the main argument to elucidate sesamoid homologies was analyzed (*Benjamin & Ralphs, 1998*; *Ponssa, Goldberg & Abdala, 2010*; *Amador et al., 2018*).

## Character evolution

The obtained data were combined in a character matrix (Supplemental Material S2), where the sesamoids were included as characters and coded as present or absent. For the reconstruction of ancestral states, data from the sesamoid matrix were optimized

**Table 1 Comparison of probabilities of occurrence of bony or cartilaginous sesamoids considering the described extant species to date in Caudata and Gymnophiona.**

| | Species described | Species reviewed | | Species with sesamoids | | Probability of occurrence | |
|---|---|---|---|---|---|---|---|
| | | Skull | Postcranium | Skull | Postcranium | Skull (%) | Postcranium (%) |
| Caudata | 757* | 200 | 147 | 16 | 0 | 2 | 0 |
| Gymnophiona | 214* | 42 | 36 | 0 | 0 | 0 | 0 |

Note:
* Data were taken from *Frost (2020)*.

onto a pruned version of the Caudata phylogeny of *Bonnet & Blair (2017)* and *Pyron & Wiens (2011)* for species that were not included in the most recent phylogeny. The tree was generated using TNT software (*Goloboff, Farris & Nixon, 2008*) and the optimization was performed with Winclada software (*Nixon, 2002*), using the default setting (unambiguous mode). Sesamoids terminology follows *Ponssa, Goldberg & Abdala (2010)* and *Abdala et al. (2019)*. We included each sesamoid in a different column of our data set (Supplemental Material S2), thus proposing their primary homology, that is, we propose that all sesamoids belonging to the same column are the same by inheritance (*de Pinna, 1991*). The proposal of primary homology is based on reasonable assessment, in this case, topology (*Agnarsson & Coddington, 2008*). However, if they fail the test of congruence and did not constitute synapomorphies on the selected cladogram, then they would not be secondary homologs (*de Pinna, 1991*). The cladogram allows us to test congruence and the hypothesis of secondary homology of the cranial sesamoid in salamanders.

## RESULTS

Three bony or cartilaginous sesamoids in Caudata were identified. One sesamoid is located on the mandibular symphysis of *Necturus beyeri* (Fig. 1A) and *Amphiuma tridactylum*. Another is a mandibular sesamoid, adjacent to the jaw articulation. The latter sesamoid is the most frequent and the one that has been most deeply studied. It is embedded in the ligament between the quadrate and prearticular bones (Figs. 2 and 3). The descending ramus of the squamosal is rectangular, flat and fused to the quadrate. The quadrate is broad; its lateral margins attach to the levator mandibulae muscles anteriorly, and posteriorly to the depressor mandibulae muscle. The ventral margin of the quadrate meets with the terminal end of the mandible. The articular bone is visible as a prominence that articulates dorsally with the quadrate, located between the prearticular and dentary (in addition to sesamoids) in some species. The bony sesamoid is located within the ligament between both the quadrate and the articular or prearticular (Fig. 2C), slightly displaced posteriorly and medially. The sesamoid is present in 12 species of seven families: *Peradactylodon persicus* (Hynobiidae) (Fig. 1D), *Cryptobranchus alleganiensis* (Cryptobranchidae), *Ambystoma jeffersonianum*, *Ambystoma mexicanum*, and *Ambystoma tigrinum* (Ambystomatidae), *Notophthalmus meridionalis*, *Notophthalmus viridescens*, and *Neurergus crocatus*, *Pleurodeles waltl* (Salamandridae), *Plethodon glutinosus* (Plethodontidae), *Necturus maculosus* (Proteidae)

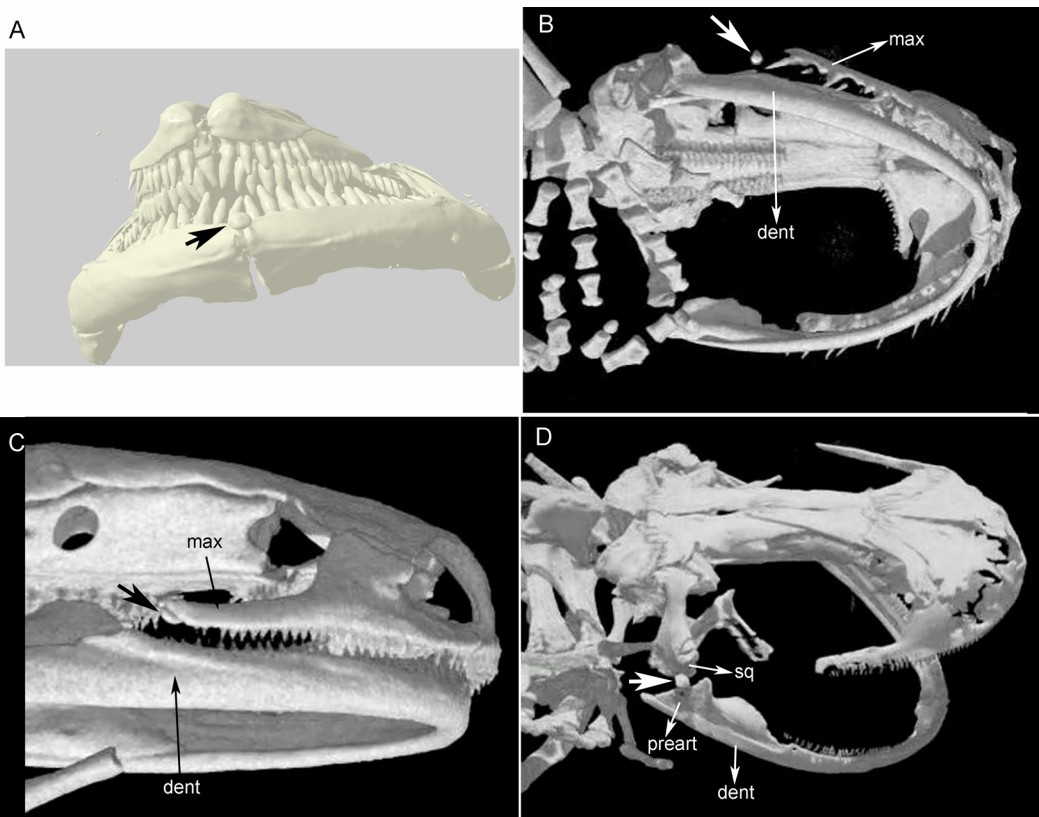

**Figure 1 Cranial sesamoids in salamanders.** Sesamoid in the mandibular symphysis of *Necturus beyeri* (UF177187), view of stl surface model of a cranial CT scan from MorphoSource, dataset DOI 10.17602/ M2/M39589 (A). Skulls of: *Hydromantes platycephalus* (uncatalogued), image credit: Digimorph.org (B), and *Karsenia koreana* (DRV5033) with a sesamoid between the posterior end of the maxilla and the lower jaw joint, image credit: Digimorph.org (C); *Peradactylodon persicus* (MVZ241494) with a sesamoid in the quadrate-prearticular joint, image credit: Digimorph.org (D). The largest arrows show the sesamoids. dent, dentary; max, maxilla; preart, prearticular; sq, squamosal.

and *Siren intermedia* (Sirenidae) (Figs. 2 and 3). The presence of this sesamoid is variable, and sometimes it is asymmetrically distributed in the same specimen, that is, being present on one side of the skull (e.g., *Ambystoma mexicanum*, FMNH22888; *Pleuredeles waltl*, FML30804). Another sesamoid was observed between the skull and the lower jaw in *Hydromantes platycephalus* and *Karsenia koreana* (Plethodontidae). It is located at the posterior end of the maxilla (Figs. 1B and 1C). Isolated structures identified as "occasional elements" are observed, which are present laterally to the terminal and subterminal phalanx of the hands in, for example, *Ambystoma mexicanum* (FMNH22888) (Fig. 4A) and *Gyrinophilus porphyriticus* (UF64645). No other cartilaginous or bony sesamoids are observed in the postcranium of the analyzed sample of urodeles. In the analyzed species of Gymnophiona, no bony or cartilaginous sesamoids are found, but occasional elements are observed surrounding the skull of some species, as in *Ichthyophis bannanicus* (MVZ236728) (Fig. 4B).

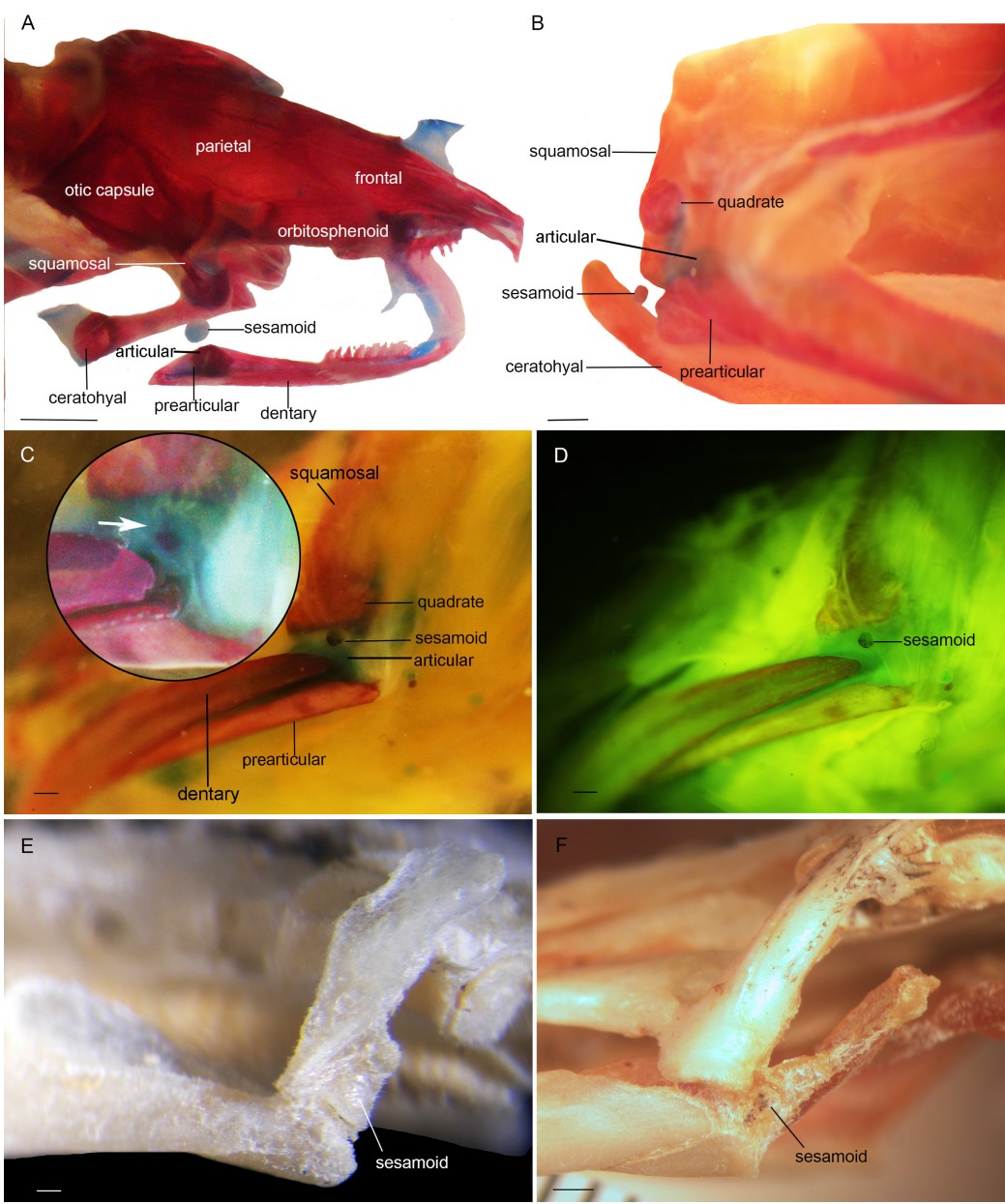

**Figure 2 Cranial sesamoids in salamanders.** Mandibular sesamoid in the quadrate-prearticular joint of the skull of *Siren intermedia* (FMNH84082) (A); *Neurergus crocatus* (FMNH19629) (B); *Ambystoma mexicanum* (FMNH22888) under white (C) and fluorescent lighting (photo credit: Jennifer Y. Lamb) (D); *Ambystoma jeffersonianum* (FMNH196112) (E); *Necturus maculosus* (FMNH21523) (F). Scale bar = 1 mm. White arrow = indicates the ligament surrounding the sesamoid in detail of the *Ambystoma mexicanum* mandibular joint.

The samples of Caudata and Gymnophiona are representative in terms of the number of species, with slightly more than 94% confidence. For a clearer indication that the absence or scarcity of cranial and postcranial bony or cartilaginous sesamoids in Caudata and Gymnophiona can indeed be considered an evidence of absence, a probability calculation

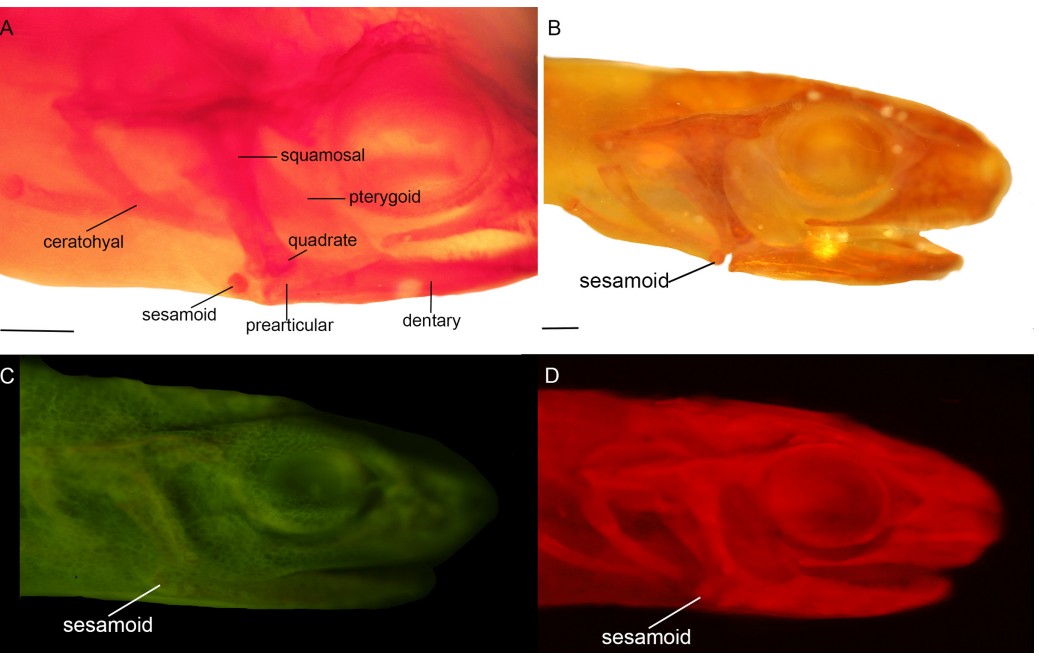

**Figure 3 Cranial sesamoids in salamanders.** Mandibular sesamoid in the quadrate-prearticular joint of the skull of *Notophthalmus viridiscens* (FMNH93537) (A); *Notophthalmus meridionalis* (FMNH93536) under white (B) and fluorescent light (photo credit: Jennifer Y. Lamb) (C and D). Scale bar = 1 mm.

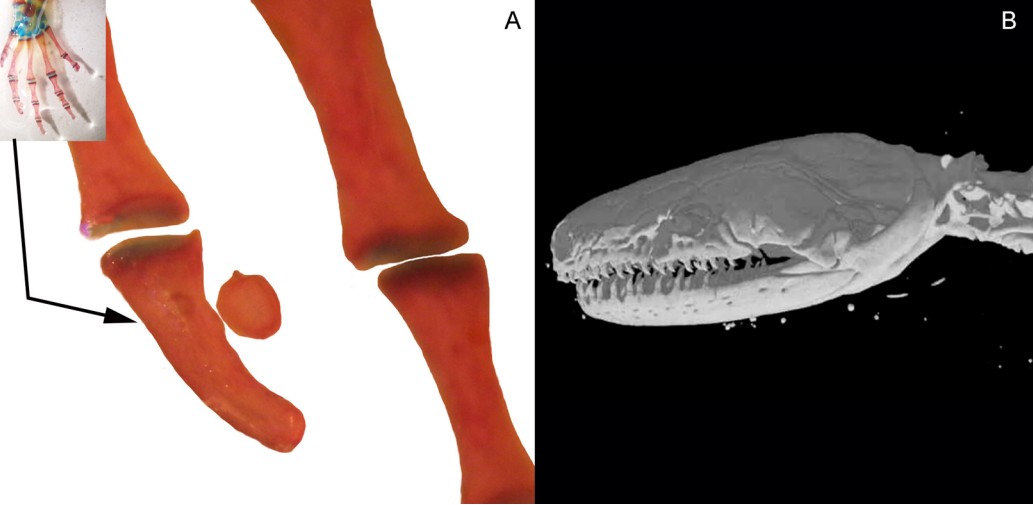

**Figure 4 Accessory structure lateral to the terminal and subterminal phalanx of hands in** *Ambystoma mexicanum* (FMNH22888) (A); and surrounding the skull of *Ichthyophis bannanicus* (MVZ236728) image credit: **Digimorph.org** (B).

based on the dataset is included. Considering the analyzed species, the probability of non-occurrences of bony or cartilaginous cranial sesamoids in Caudata is very high (98%, see Table 1). On the other hand, the probability of non-occurrence of bony or

cartilaginous postcranial sesamoids in Caudata and of bony or cartilaginous cranial and postcranial sesamoids in Gymnophiona is 100% (Table 1).

## Sesamoid evolution

The survey resulted in a very representative dataset, since all nine families of Caudata were studied. Optimization of the sesamoid in the mandibular symphysis shows that it is a convergent acquisition (Fig. 5). The mapping analyses of the mandibular sesamoid dataset onto the phylogeny reveal its convergent appearance in seven families (Fig. 5), thus rejecting the secondary homology proposal. The optimization was ambiguous at the base of the clades of the *Notophthalmus* species and at (*Ambystoma tigrinum* + *A. ordinarium*). In the *Hydromantes platycephalus* + *Karsenia koreana* (Plethodontidae) clade, the presence of the sesamoid between the maxilla and the lower jaw was optimized as a putative synapomorphy (Fig. 5).

## DISCUSSION

Our survey of sesamoids in Caudata indicates that the only three bony or cartilaginous sesamoids present in the group are associated with the mandible: one with the symphysis, another with the quadrate-articular joint of the skull, and another between the maxilla and the lower jaw. As far as we know, this report provides the first record of sesamoids in the mandibular symphysis and at the posterior end of the maxilla of tetrapods. No sesamoids were observed in the postcranial joints. Furthermore, in Gymnophiona, no bony or cartilaginous sesamoids were observed.

The very conspicuous sesamoid in the mandibular symphysis suggests some kind of particular mechanical stress acting on the mandible of *Necturus beyeri* and *Amphiuma tridactylum*, although its presence in only these two taxa makes the ecological or biomechanical inferences excessively speculative. On the contrary, the sesamoid located between the maxilla and the lower jaw is a new synapomorphy that supports the node of *Hydromantes platycephalus* and *Karsenia coreana*.

The pool of species reviewed in this study showed that the mandibular sesamoid is present in seven of the nine families included in the order. However, that sesamoid is infrequent at the specific level, since it is present in only 16 of the 212 examined species (the order includes 757 recognized species; *Frost, 2020*). The skulls of some species in which sesamoids were observed were previously described, but those descriptions included no reference to sesamoids; for example, *Siren intermedia* (*Reilly & Altig, 1996*), *Ambystoma tigrinum* (*Naylor, 1978*; *Pedersen, 1993*; *Reilly & Lauder, 1990*), *Notophthalmus viridiscens* (*Naylor, 1978*; *Reilly, 1986*); *Pleurodeles waltl* (*Corsin, 1966*). *Buckley, Wake & Wake (2010)* described the skull of *Karsenia koreana* based on the same specimen reviewed in this work (DRV5033). However, they did not mention the sesamoid between the maxilla and the lower jaw, although it is very conspicuous in their illustrations (see fig. 2A–2D of *Buckley, Wake & Wake (2010)*).

The optimization of the mandibular sesamoid reveals its convergent appearance in the different families; thus, the secondary homology proposal is rejected. According to our

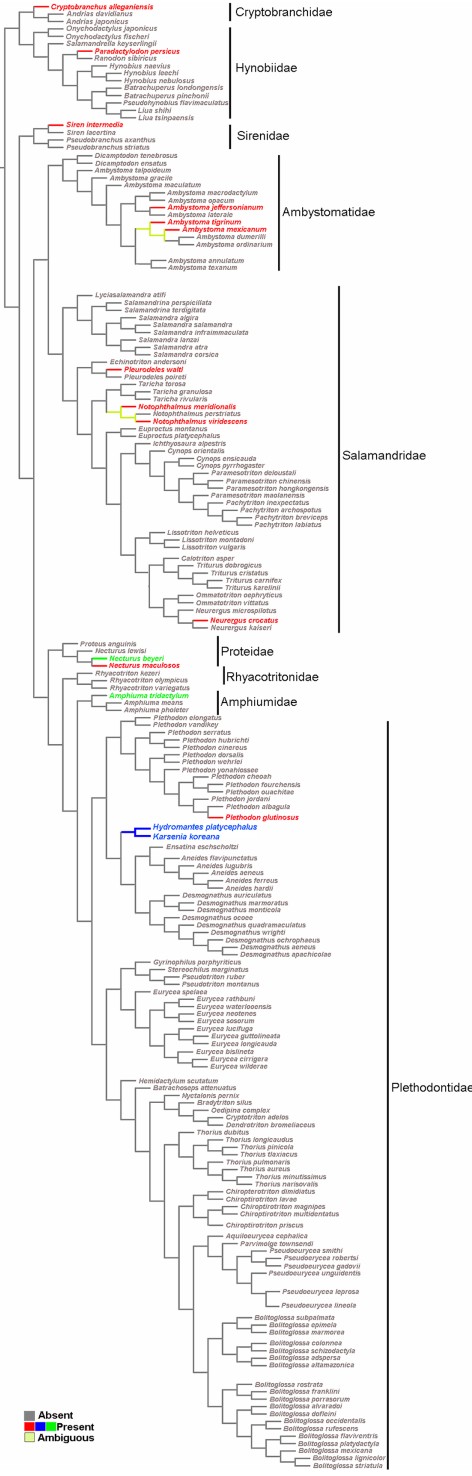

**Figure 5 Salamander phylogeny used in the optimization analysis of mandibular sesamoid characters follows relationships proposed by *Bonnet & Blair (2017)*, and *Pyron & Wiens (2011)* for species not included in the most recent phylogeny.** Mapping of sesamoid of the mandibular symphysis is indicated in green; mapping of mandibular sesamoid, located between the quadrate and prearticular bone is indicated in red; mapping of sesamoid between the skull and the lower jaw, located at the posterior end of the maxilla is indicated in blue.

optimization, the absence of this sesamoid is plesiomorphic to Caudata. Three and 15 species of the most basal families, Cryptobranchidae (including four species; *Frost, 2020*) and Hynobiidae (including 81 species; *Frost, 2020*), respectively, were reviewed and one species with a bony or cartilaginous mandibular sesamoid was recorded in each family. The descriptions of 12 Caudata fossil species were reviewed, and none of them mentioned the presence of sesamoids. The stem-group of salamanders occurred during the Middle Jurassic, and was represented by species such as *Karaurus sharovi* (Karauridae), *Kokartus honoraius* (Karauridae) and *Marmorerpeton kermaki* (Incertae sedis) (*Ivakhnenko, 1978*; *Evans, Milner & Musset, 1988*; *Gao & Shubin, 2003*; *Carroll, 2007*). These specimens and the first crown-group of Caudata, represented by *Chunerpeton* (Cryptobranchidae) of the Middle Jurassic and *Beiyanerpeton* and *Qinglong triton* (Salamandroidea) of the Late Jurassic (*Gao & Shubin, 2012*; *Gao, Chen & Jia, 2013*; *Jia & Gao, 2016*), are remarkably well-preserved fossils and do not present sesamoids. Even when more hynobiidae should be sampled, the available evidence of the basal clade Hynobiidae + Cryptobranchidae and the fossil record support the hypothesis of the absence of sesamoids as a plesiomorphic state in Caudata.

We propose the mandibular sesamoid of Caudata as a primary homology to that of anurans. However, in this group, the sesamoid has only been reported in *Barbourula busuangensis* (*Clarke, 1987*; *Roček et al., 2016*) and *Chacophrys pierotti* (*Fabrezi, Goldberg & Chuliver Pereyra, 2017*) and, based on our findings, now in *Alytes obstetricans* (Alytidae, see https://www.morphosource.org/Detail/MediaDeandtail/Show/media_id/13730), *Kassinula witei* (Hyperoliidae, see https://www.morphosource.org/Detail/MediaDetail/Show/media_id/21067) *Silvertoneia flotator* (Dendrobatidae, see https://www.morphosource.org/Detail/MediaDetail/Show/media_id/63791); *Barycholos pulquer* (Craugastoridae, see https://www.morphosource.org/Detail/MediaDetail/Show/media_id/14710); *Ansonia mcgregori* (Bufonidae, see https://www.morphosource.org/Detail/MediaDetail/Show/media_id/14201). It is likely to be present in a wider sample of anurans and might be considered present in the developmental program of the group. In *Barbourula busuangensis*, the sesamoid was described in the ligament, passing from the squamosal shaft to the posterior end of the angular bone of the mandible (*Clarke, 1987*; *Roček et al., 2016*). According to its topology and related ligament, in salamanders and anurans the sesamoid would meet the proposed primary homology criterion. However, a wider taxon sampling is needed to test the secondary homology of this sesamoid in Caudata and Anura.

Three groups of hypotheses about the origin of frogs, salamanders and caecilians persist in the literature: the "temnospondyl hypothesis", the "lepospondyl hypothesis" and the "polyphyly hypothesis", of which the former was the most widely supported for a long time (*Ruta & Coates, 2007*). However, recent re-analyses strongly support the lepospondyl origin of lissamphibians (*Marjanović & Laurin, 2013*, *2019*). Whatever the affinities of the three modern orders with Paleozoic amphibians, the records of temnospondyls and lepospondyls do not include sesamoids (*Walsh, 1987*; *Gardner, 2003*, *Carroll, 2007*; *Fortuny et al., 2016*; see species in Supplemental Material S1). Further research is needed

to find out whether fibrocartilaginous sesamoids are absent in these groups as well. Sesamoids are identified "in relation to" other anatomical elements, such as ligaments, tendons, and joints (*Fontanarrosa, Fratani & Vera, 2020*). In disarticulated fossil remains sesamoids are challenging to identify, which might explain their report in association with postcranial joints in only two fossils of the anuran *Xenopus* (*Henrici & Baéz, 2001*; *Wang, Jones & Evans, 2007*).

The proposed functions of sesamoids include protecting tendons against friction, compression, tension, or injuries (*Sarin et al., 1999*; *Otero & Hoyos, 2013*; *Regnault, Hutchinson & Jones, 2016*; *Abdala, Vera & Ponssa, 2017*; *Zhang et al., 2017*). The integration of the mandibular sesamoids with other elements of the mandibular joint (bones, muscles, tendons and ligaments) would be crucial to generate the strength for feeding through complex and coordinated movements. Interestingly, it is not possible to propose a direct relationship between dietary selection or feeding habits and the presence of a mandibular sesamoid. Species with a mandibular sesamoid have a similar diet to that of species without sesamoids: they are predators with a wide range of prey types and sizes, from macroinvertebrates to small vertebrates, including annelids, mollusks, ants, spiders, centipedes, snails, sowbugs, beetles, mayflies, stoneflies, fish, amphibians, etc. (*Hamilton, 1932*; *Bardwell, Ritzi & Parkhurst, 2007*). In a physiological position, the sesamoid fits between the squamosal and the prearticular, and is located among the jaw bones instead of covering the joint, the most common position of sesamoids. More comparative data is necessary to test the hypothesis of biomechanical restrictions linked to feeding and development of the mandibular sesamoids.

We prefer not to include among sesamoids the occasional elements found in of *Ambystoma mexicanum*, *Gyrinophilus porphyriticus* and *Ichthyophis bannanicus*, among others, due to their somewhat anecdotal record. Thus, we believe they should be considered occasional structures (see also *Kunc et al., 2020*) until their location embedded in a tendon is confirmed with histological studies.

The absence of bony or cartilaginous postcranial sesamoids in salamanders is striking and would be another particular characteristic of their limbs. Limbs of salamanders have a unique development among extant tetrapods, since they are characterized by the absence of the apical ectodermal ridge and the formation of the distal mesopodial and autopodial elements without a continuous condensation linking them to more proximal cartilages (*Shubin & Alberch, 1986*; *Franssen et al., 2005*; *Fröbisch & Shubin, 2011*; *Kumar et al., 2015*; *Kearney, Hanken & Blackburn, 2018*). In the context of limb development of urodeles, several morphological patterns resulting from heterochrony— timing differences in development—have been described (*Wake, Wake & Wake, 1983*; *Blanco & Alberch, 1992*; *Shubin, 2002*). These patterns consist of variations in the stages at which the fore and hind limbs develop (*Smith, 1912*; *Hanken, 1982*; *Babcock & Blais, 2001*; *Nye et al., 2003*; *Bininda-Emonds et al., 2007*), or in the sequences in which several skeletal elements develop in the limbs (*Shubin & Alberch, 1986*). In the absence of the AER, the small plethodontid *Bolitoglossa occidentalis* lacks phalangeal elements, which is interpreted as a paedomorphic character (*Alberch & Alberch, 1981*). Sesamoids develop

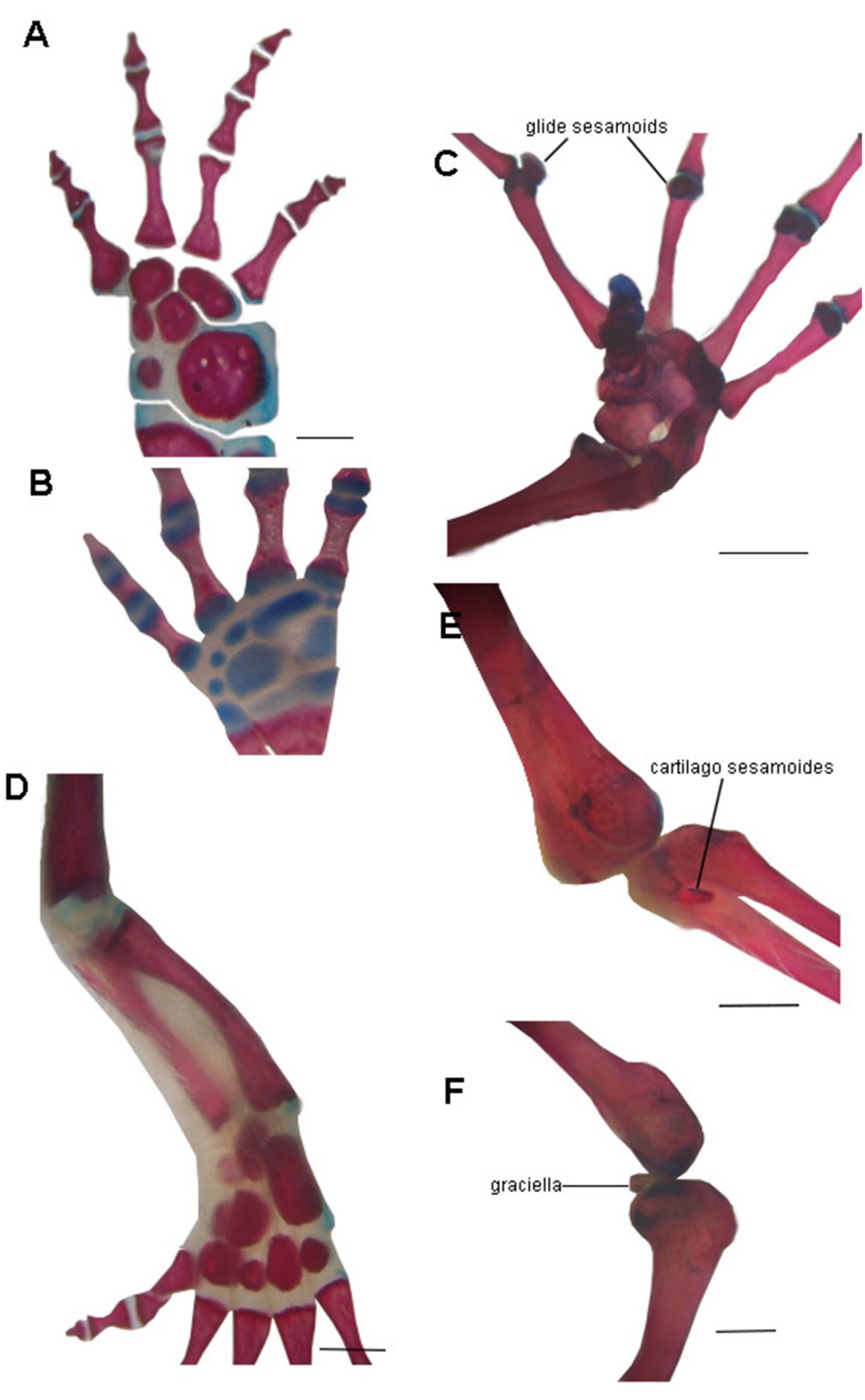

**Figure 6 Examples of joints in salamanders and frogs showing the difference in the complexity of the joint surfaces.** (A) Manus of the facultative paedomorphic species *Pleurodeles waltl* (FML30803) (B) Manus of the paedomorphic species *Siren intermedia* (FMNH84082) (C) Manus of the anuran species *Leptodactylus latinasus* (FML30832), showing the glide sesamoids in the inter-phalangeal joints. (D) Hindlimb of *Pleurodeles waltl* (FML30803), the simple articular surfaces of the knee and ankle joints are evident. (E) Ankle and (F) Knee joints of *Leptodactylus bufonius* (FML30833) where the related sesamoids are visible. Scale bar = 1 mm.

later than the other skeletal elements of anuran limbs (*Ponssa, Goldberg & Abdala, 2010*; *Vera & Ponssa, 2014*; *Vera, Ponssa & Abdala, 2015*); thus, their absence in the salamander clade (both in large and small species), can be speculated to be a paedomorphic pattern, as a consequence of a truncated development. Whether this pattern is produced by the absence of the AER or any other changes in developmental mechanisms remains to be investigated. Interestingly, the limb joints of urodeles are also highly paedomorphic. The articular surfaces of the long limb bones are rather simple, remain cartilaginous, and lack secondary ossification centers (*Haines, 1942*; *Meng et al., 2019*) and the complexities present in the concave bony epiphyses of other tetrapods (see for example, *Hanken, 1982*, Fig. 4B) (Fig. 6). These simple structures might explain the lack of bony or cartilaginous sesamoids, since the tendons would not be subjected to the pressure and effort of sliding on a hard and sculpted surface.

There are no records of postcranial sesamoids in salamanders, caecilians, branchiosaurids, or other lepospondyls and temnospondyls (*Walsh, 1987*; *Carroll, 2007*; *Fortuny et al., 2016*; *Gardner, 2003*; *Gruntmejer, Konietzko-Meier & Bodzioch, 2016*). An excellent fossil record of Paleozoic branchiosaurids presents a clear view of the limb ossification pattern in the Temnospondyli *Apateon* (*Fröbisch, Carroll & Schoch, 2007*), which lacks sesamoids. The process of limb ossification in *Apateon* is similar to that of urodeles in some aspects, such as the preaxial dominance in limb development or the limb ossification sequence, which has been discussed as a critical character to elucidate the relationship of these taxa (*Fröbisch, Carroll & Schoch, 2007*; *Fröbisch, 2008*). When plotted on the hypothesis of lissamphibian relationships to basal tetrapods, postcranial sesamoids require two steps to achieve a convergent evolution with postcranial sesamoids present in frogs and amniotes (Fig. 7). The absence of postcranial sesamoids results in the plesiomorphic state in tetrapods.

Caecilians are elongated, snake-like amphibians, completely lacking limbs and girdles. They have a terrestrial, surface-cryptic, or burrowing lifestyle, except for Typhlonectidae, whose members are secondary aquatic or semi-aquatic (*Taylor, 1968*; *Estes, 1981*; *Wilkinson & Nussbaum, 1999*). The lack of postcranial sesamoids could be a logical consequence of the lack of postcranial joints of the crown group Gymnophiona. However, in the two putative stem-groups of caecilians, *Eocaecilia micropodia* from the Lower Jurassic, which has reduced limbs (*Jenkins & Walsh, 1993*), and *Rubricacaecilia monbaroni* from the Lower Cretaceous, to which a left femur showing a trochanteric crest has been tentatively attributed (*Evans & Sigogneau-Russel, 2001*), no sesamoids have been reported. According to the dynamic model proposed in *Abdala et al. (2019)*, sesamoids can

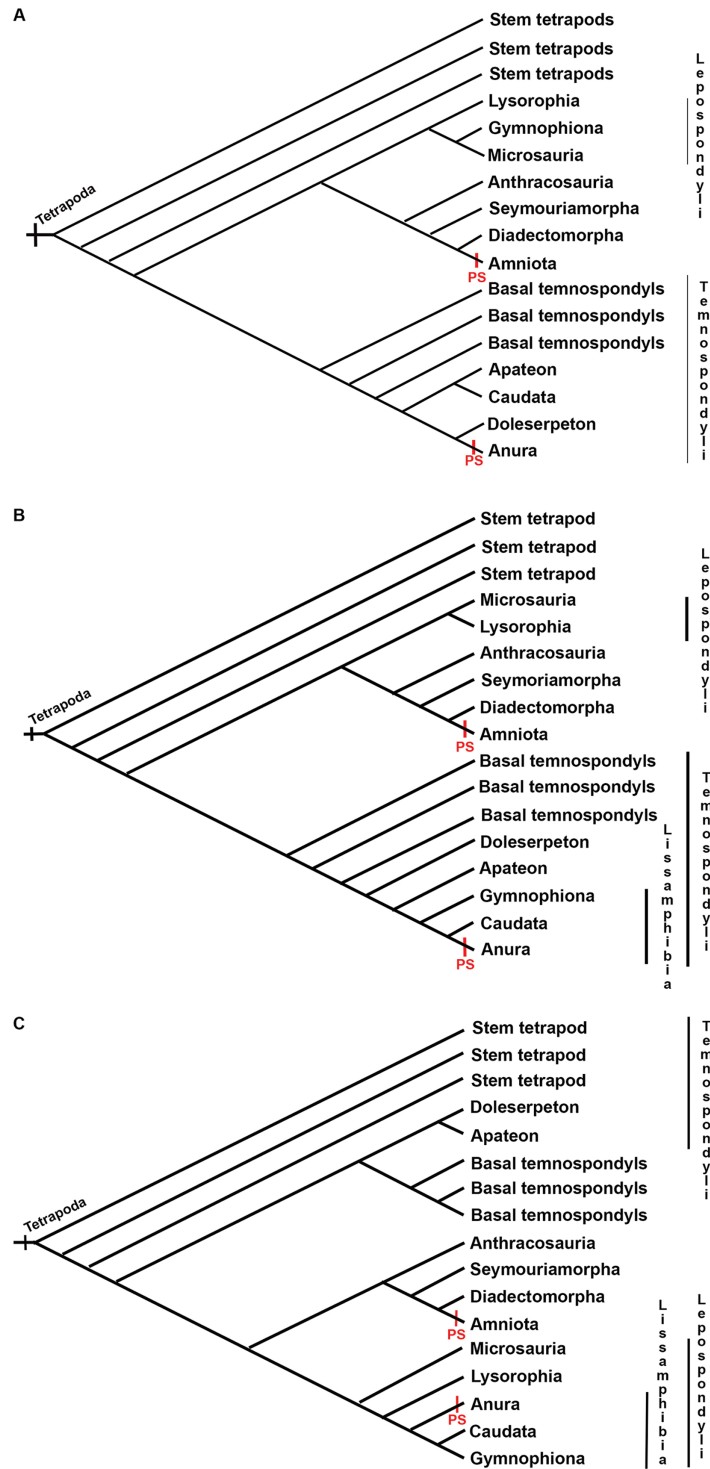

**Figure 7 Three discussed hypotheses of interrelationships of the three modern amphibian groups and their relationships to possible Paleozoic records.** (A) Polyphyly hypothesis simplified from *Ruta & Coates (2007)*; (B) temnospondyl hypothesis (*Trueb & Cloutier, 1991*; *Milner, 1993*; *Ruta, Coates & Quicke, 2003*); (C) lepospondyl hypothesis (*Laurin & Reisz, 1997*; *Laurin, 1998*; *Vallin & Laurin, 2004*). Red bars indicate the presence of postcranial sesamoids (PS) as has been known until present.

attach to and detach from the long bone epiphyses and diaphyses during both ontogeny and phylogeny (see also *Parsons, 1904*; *Manners-Smith, 1908*; *Barnett & Lewis, 1958*; *Cheong et al., 2017*, among many others). Thus, the observed trochanteric crest could be interpreted as an attached sesamoid. New fossil evidence could fill the information gap about sesamoid evolution in this group.

Since caecilians are limbless and mainly fossorial animals, the diversification of their cranium might have been conditioned by the functional demands of head burrowing (*Sherratt et al., 2014*). This fossorial mode of life might have been the most substantial influence on the parietal and quadrate-squamosal modules (*Bardua et al., 2019*). This strong integration of the skull bones could explain the lack of sesamoids in the quadrate-squamosal module. Moreover, the rearrangement of jaw-closing muscles across Gymnophiona influences the jaw joint (*Bardua et al., 2019*). The involved tendons would not be subjected to the necessary tension threshold for the development and evolution of sesamoids in this group.

The tissue composition of sesamoids can be diverse and can include cartilage, bone, or fibrocartilage. In the latter case, histological samples would be necessary to corroborate its presence or absence in salamanders and caecilians.

## CONCLUSIONS

Our data show the presence of three cranial bony sesamoids in Caudata, which is remarkable considering that cranial sesamoids are rare among tetrapods. Our data also indicate that Caudata lacks bony or cartilaginous postcranial sesamoids, whereas Gymnophiona lacks bony or cartilaginous cranial and postcranial sesamoids. Our ancestral state reconstruction indicates that the plesiomorphic state to the Caudata is the absence of bony or cartilaginous cranial sesamoids.

## ACKNOWLEDGEMENTS

We thank Santiago Nenda (MACN) for providing study material, G. Fontanarrosa for her comments on the manuscript, and S. Nannis for her help with English. M.L.P is very grateful to Alan Resetar and Joshua Mata for their hospitality in the herpetology collection of the FMNH. M.L.P. We thank Jennifer Y. Lamb for the fluorescent light photos. Morphosource.org and DigiMorph.org, and Ambibia Tree authorized the use of images. We are also thankful to J. M. Hoyos (Pontificia Universidad Javeriana, Colombia), and two anonymous reviewers for their help in improving our work.

### Funding

This work was supported AGENCIA NACIONAL DE PROMOCIÓN CIENTÍFICA Y TECNOLÓGICA (Préstamo BID PICT 2015/1618 to María Laura Ponssa, PICT 2016-2772 and PICT-2018-00832 to Virginia Abdala), Consejo Nacional de Investigaciones Científicas y Técnicas (PIP CONICET 389 to Virginia Abdala) and Field Museum of

Natural History (grant to María Laura Ponssa). The funders had no role in study design, data collection and analysis, decision to publish, or preparation of the manuscript.

## Grant Disclosures
The following grant information was disclosed by the authors:
AGENCIA NACIONAL DE PROMOCIÓN CIENTÍFICA Y TECNOLÓGICA: PICT 2015/1618, PICT 2016-2772 and PICT-2018-00832.
Consejo Nacional de Investigaciones Científicas y Técnicas: PIP CONICET 389.
Field Museum of Natural History.

## Competing Interests
Virginia Abdala is an Academic Editor for PeerJ.

## Author Contributions
- María Laura Ponssa conceived and designed the experiments, performed the experiments, analyzed the data, prepared figures and/or tables, authored or reviewed drafts of the paper, and approved the final draft.
- Virginia Abdala conceived and designed the experiments, performed the experiments, authored or reviewed drafts of the paper, and approved the final draft.

## Animal Ethics
The following information was supplied relating to ethical approvals (i.e., approving body and any reference numbers):
   We worked with material deposited in scientific collections, not with live animals.

## Data Availability
   Raw measurements are available in the Supplemental Files.

## Supplemental Information
Supplemental information for this article can be found online at http://dx.doi.org/10.7717/peerj.10595#supplemental-information.

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
