# Peer review of "Sesamoids in Caudata and Gymnophiona (Lissamphibia): absences and evidence"

_PeerJ, doi:10.7717/peerj.10595_

## Round 0.1 · original submission · Major Revisions

Three reviewers have shared comments and broadly agree that the issue of homology needs to be more deeply addressed (e.g. test of congruence vs. phylogeny), among other important issues; but as do I, they all agree there are interesting issues involved, too. This will need re-review so please detail all issues in a point-by-point response, carefully. Thank you.

Reviewer 1 ·

Basic reporting

The manuscript Evolution of sesamoid bones in Caudata and Gymnophiona (Lissamphibia) presents results about the presence/absence of sesamoids in selected extant amphibians.
I do not agree with the presence of sesamoids in those groups is understudied since their morphology is well known and sesamoids are not frequent or/and remarkable in those groups.
The goals are clear and unambiguous but the title of the manuscript biases the reader in the wrong direction since the exploration of a large sample has few results in order to discuss and/or interpret the evolution of these structures.
The introduction and background are disorganized to place the focus of the research. As the definition of sesamoid contains many aspects (gene expression, tissues properties, movements, ontogeny, topology) it could be necessary to explain what kind of findings the authors are expecting in Urodela and Gymnophiona (for example, sesamoid associated with skull articulations) as the most important issue and secondarily, appendicular sesamoids in Caudata.
Reports of sesamoids in the anuran skull are not frequent. The squamosal-mandibular sesamoid in Bombinatoridae (Clarke, 1987) needs to be checked because the figure of the dry skeleton in that paper lacks lower jaw The pieces attributed to the sesamoid could be part of the lower jaw. Another report of a small sesamoid in the tendon of muscle levatorae mandibulae externus is described as rare sesamoid in the lower jaw articulation of a neobatrachian frog (Fig. 16. In Fabrezi et al, 2017, J. Exp. Zool B, 328: 546–57).
About the presence of sesamoids in postaxial and appendicular skeletons in Urodela, it is necessary to justify the exploration beyond the absence of literature data.
I find valuable novel data about the variation in the presence of sesamoids in some Caudata but the discussion is based on many ideas that are not linked with few supporting results.

Experimental design

The study presents data about the presence or absence of sesamoid in cleared and stained whole mounts and dry skeletons. However, both anatomical preparations present limitations depending on the preserved specimen (decalcification for long exposition to fixative liquid, age of the individual), or loss of bones with soft tissues. Furthermore, it is necessary to provide the location of the sesamoid within a ligament because there are different muscles that open and close the mouth in the mandibular joint. Similar information should be provided for sesamoids in limbs.

Validity of the findings

Few results (observed variation in Caudata) limit the discussion. The absence of sesamoid precludes any evolutionary interpretation for Gymnophiona.
Discussion about the presence of mandibular sesamoid in some species of Ambystomatidae, Salamandridae, Plethodontidae, Proteidae and Sirenidae should be related to results that are novel contributions and opposed to features that are already known and/or not derived from results.
Diet, biomechanical interpretations, and fossil data to discuss the mandibular sesamoid and early limb development (ectodermal apical ridge) to discuss appendicular sesamoids are not providing developmental/ecological/evolutionary arguments as the title suggests.

Additional comments

The authors must consider that morphological variation and morphological evolution are not synonymous.

Reviewer 2 ·

Basic reporting

Basic Reporting
The body of the text and approach lacks a coherent series of tests and criteria necessary to evaluate homology of sesamoids, particularly between amphibian ‘mandibular sesamoids’ and those sesamoids found in other lineages of tetrapods. The illustrations and photographs do not offer enough anatomical details for the reader to adequate assess the assertions the authors make regarding the positions, composition, and topology of the sesamoids under review. The submitted manuscript requires more attention to some aspects of the English. There are several instances of grammatical errors (e.g., pronouns) that affect the interpretation of the anatomical descriptions and interpretations.

Experimental design

Experimental Design
It is unclear what criteria the authors are using to define what a sesamoid is. Descriptions and prose suggest they occur in any number of tissues such as ligament AND muscle as well as anywhere else, but other accepted definitions of sesamoids (sensu Benjamin & McGonagle https://doi.org/10.1007/978-1-4419-0298-6_4,; Bengamin and Ralphs https://doi.org/10.1046/j.1469-7580.1998.19340481.x) are quite precise in that sesamoids are intratendinous nodules that often have trochlear systems associated with them, all of which arise epigenetically due to compressive forces acting about wrapping muscle tissues. Thus, describing the ‘mandibular sesamoid’ of amphibians as lying within a ‘ligament’ between the quadrate and prearticular bones goes against the above definition. Arguments regarding these conflicting definitions are welcome, however the authors here do not provide clarity or precision through descriptive text or figures to this purpose. For example, the presence of a nodule within ‘ligament’ within the jaw joint in the amphibian taxa discussed here suggests that an acceptable, and testable hypothesis is that this is a mineralization of Meckel’s cartilage or a rudiment of an ‘articular’ bone, rather than an intramuscular nodule (or sesamoid sensu Haines, Benjamin & Ralphs). But the authors do not provide the readers any evidence to evaluate their case, whether it be histology of the mineralized tissues and their surroundings, imaging showing soft tissue topologies or other data. The photos provided are interesting, but lack clear positional data and magnification for the readers to truly see the work.
This lack of clarity extends into the authors interpretations of function and homology in the Discussion. They seem to homologize all ‘sesamoids’ in the adductor chamber of reptiles as ‘mandibular sesamoids’ sensu the structure in the amphibian jaw joint. But the cartilagos transiliens of turtles and crocodiles have no associations with the jaw joint (i.e., quadratoarticular joint) and have clearly been shown to be intramuscular/intratendinous sesamoids, rather than intercalary, ligamentous, articular nodules akin to the amphibian ‘mandibular sesamoid’. They go on to suggest the amphibian mandibular sesamoid serves a jaw locking function akin to the crocodilian pterygoid flange & sesamoid system. But they do not offer any description of the biomechanical structures and phenomena that facilitate this behavior. I think in general, if these authors refer to the work and exposition of Montero et al., 2017 study of nodules in the otic joint of reptiles, they will find an example of a rich and concise exploration of sesamoids, and sesamoid-like structures in at least cranial skeletal tissues.
Finally, I am unable to evaluate the distribution of sesamoids in the limbs of amphibians, but I suggest the authors take care in their comparisons and homologies between appendicular and cranial sesamoids and sesamoid-like structures.

Validity of the findings

Validity of Findings
As discussed above, it is challenging to evaluate the validity of findings about phylogeny or function for example, because the authors do not offer enough precise details of morphology in their system of interest or clarity of hypotheses of homology being tested.

Additional comments

Overall, new data on the diversity and distribution and tissue biology of sesamoids are very much. More clarity on the distribution and diversity of ‘sesamoids’ is welcome in vertebrate evolution and skeletal tissue biology. However, several fundamental issues with the approach in this paper hamper its effectiveness. It has potential to be a welcome contribution, but in its current form, it lacks the evidence and exposition necessary for robusticit in the scientific literature.

·

Basic reporting

1. ¿Is the argument logical?

Yes

2. Are the methods suitable and results plausible?

Yes

3. Are the findings adequately described and discussed?

Yes

4. Is the interpretation of the data appropriate in light of available theory?

Yes

4. Are key papers in the field cited?

Yes.

The authors respond to the quiestion posed in the paper. However, this is an atypical paper, because it is a good example of what we could call as "negative" results since they did not find what they were looking for, which makes it very interesting because it shows that we have to do enough research to be able to draw powerful conclusions. Hypotheses scientific research are usually raised to search for “positive” results, and if the opposite is found, it is considered that it is not worth presenting them to the community. In this case, the authors dare to do it, which is very important for us, researchers in vertebrate morphology, because they save us an important step in the search, in this case, for sesamoid elements in Caudata and Gymnophiona. This being the case, it is difficult to evaluate this paper: what to say about something that does not exist, as in the case of the Cecilias, or that is in very few, as in the case of salamanders? In this situation, the authors take the only path left to them, and that is to find out what is the probable evolution of these structures in current and fossil groups, including even the other groups of tetrapods, all this using the bibliography they have by hand.

Experimental design

• There is not experimental design. I suppose the tha authors used material already cleared and stained because they do not say anything about the protocol they used for these.

Validity of the findings

• Considering the amount of species and specimens they observed, I think their findings are valid.

Additional comments

I have some comments:

1. About the references: Transfer the reference from line 416 to line 411.
Change Vera et al. 2013 to Vera et al. 2014, or viceversa

2. About the homology:

The authors claimed that (lines 300-302) “Under this premise, we propose the primary homology of most mandibular sesamoids in tetrapods, since all of them are embedded or related to the ligaments linking the skull and the mandible”.

But, if they proposed a topology (phylogeny?) including their results, why do they keep calling these relationships as primary homology? This one is based on position and similarity of forms, prior to the phylogenetic analysis, but when characters are already included as part of a phylogeny, they should be called secondary homologies. In fact, it is clear that the trees show that these sesamoids would not be homologous since they appear in trees that the taxa have polyphyletic relationships, at least among the salamander species studied.

3. Because the conclusions must be inferred from the discussion, it would be worthwhile leaving them rather explicitly at the end of the paper.

---

## Round 0.2 · Major Revisions

I am sorry but in re-reading the reviews and the Rebuttal, I feel that to better satisfy the reviewers 2 things are needed for the MS to be better suited to sending back for re-review (and I'll do that swiftly once resubmitted):

1. Provide the full data matrix used to do the optimization analysis, and a clearer detail in the Methods how the optimization was done (was ACCTRAN or DELTRAN optimization used, or neither?). This is absolutely critical so that the study is reproducible/re-usable/etc. as per journal criteria.

2. As homology is still a keyword in the MS, please make it clearer how primary vs. secondary homology is being used in the study. Some readers might not be immediately familiar with these terms and why one or the other is/is not evolutionary.

I recommend that another pass at proofreading the MS is done. On a read through, I felt that it still is awkward to read and sometimes the meaning of key phrases is unclear.

I hope you understand my decision here is purely meant to help improve the MS and make the peer review process easier for you and the reviewers.

---

## Round 0.3 · Minor Revisions

While the reviewer calls for "minor" revisions, there remain substantial issues of dispute and there is still a need for refinement of the English to render it a smoother read for our audience. Please address the recommended changes in detail, including how the readability has been addressed. Thank you.

Reviewer 2 ·

Basic reporting

Please find all of my comments in one section

Experimental design

Please find all of my comments in one section

Validity of the findings

Please find all of my comments in one section

Additional comments

Dear Authors,
Thank you for your careful inclusion of comments and your discussion and clarification of patterns of sesamoids in the heads of amphibians. I remain confident that this submission is a worthy contribution. Please excuse my delay in the review.

Perhaps it is a philosophical difference in approach, but I remain skeptical that it is correct or wise to synonymize all sesamoids in the head as the same sesamoid (i.e., the mandibular sesamoid). Given than many of the sesamoids presented here lie on the jaw opening side of the jaw joint, rather than the jaw closing side of the head (where other sesamoids can occur), these sesamoids differ greatly in their topological position and developmental position (e.g., pharyngeal arch 1 vs 2) compared to sesamoids in the jaw adductor muscles. This alone suggests nonhomology. I also disagree with employing Diogo et al’s 2015 overly broad position on what serves as homology, particularly in this study on structures in the head. Applying findings from limbs, which develop from arguably a more simple anlagen of layers, to that of the tetrapod skull, which has numerous different cell populations (e.g., neural crest), ossification patterns (membrane vs endochondral) and quite specific domains of differently-innervated musculature, seems overly superficial and lacks the attention to detail many other authors have clarified. Even in the limbs, I cannot fathom how sesamoids on the flexor side of a limb could possibly be considered 'the same' as sesamoids on the extensor side of a limb. I do agree that the identity of some neighboring bony structures may not always be significant compared to the position of sesamoids in clearly homologous muscles, for example the arrangement of post-dentary elements in different lineages of tetrapods is somewhat independent of sesamoids in the Bodenaponeurosis of Sauropsids. Though I wholly disagree with the finding of Montero et al 2017 that the surangular bone has anything to do with the jaw muscle, sesamoid, tendon relationships in crocodilians, but that is only part of our conversation here and not part of the paper.


Ln 292: these sesamoids are not ‘within jaw bones’ they are between or among them. ‘Within’ would mean they were somehow inside the elements.

Here it is important to say that there remains a number of errors in the writing that can greatly affect description and interpretation of position. I absolutely understand the authors’ challenge in the expectation of writing in English for these journals and I appreciate my privilege in not having this challenge. I certainly could not write a paper in Spanish that is remotely as clear as this one, even if I still see issues with clarity. I leave this up to the editor to help shepherd.



Ln 246 The discovery that ‘sesamoids’ may merge with bones plastically, or developmentally/evolutionarily while under different mechanical loads has a much deeper history than Abdala’s 2yr old paper. Please see Barnett and Lewis 1958 and references therein about traction epiphyses and apophyses. https://www.ncbi.nlm.nih.gov/pmc/articles/PMC1244993/?page=10.
There is also a vibrant literature on traction epiphyses in the modern medical and orthopedic literature.

---

## Round 0.4 · accepted · Accept

Well done - there may be lingering conceptual disagreements but the science has improved in terms of its clarity and in closer agreement with the reviewer on some points; and the writing has been smoothed out. There is no merit in further review at this stage, out of respect for reviewers' time, and the paper now seems a worthy contribution to the literature. Congratulations!